# COVID-19 Vaccination Might Induce Reversible Cerebral Vasoconstriction Syndrome Attacks: A Case Report

**DOI:** 10.3390/vaccines10050823

**Published:** 2022-05-23

**Authors:** Anne Marie Lund, Mohammad Al-Mahdi Al-Karagholi

**Affiliations:** 1Faculty of Health and Medical Sciences, University of Copenhagen, 2200 Copenhagen, Denmark; mohammad.al-mahdi.al-karagholi@regionh.dk; 2Department of Neurology, Rigshospitalet Glostrup, 2600 Glostrup, Denmark; 3Department of Neurology, Nordsjaellands Hospital—Hilleroed, 3400 Hilleroed, Denmark

**Keywords:** corona virus, Pfizer, vaccination, headache

## Abstract

A 30-year-old male diagnosed three years previously with reversible cerebral vasoconstriction syndrome (RCVS) presented to the department of neurology with an accumulation of attacks mimicking previous RCVS attacks and fulfilling the diagnostic criteria for RCVS after receiving the first Pfizer COVID-19 vaccine. The neurologic exam, blood samples, electrocardiogram (ECG), and computer tomography of the head (CTC) were normal. The patient was treated with the angiotensin 2 receptor antagonist, losartan, with a good response and was discharged with a prescription for losartan lasting until three days after the second Pfizer COVID-19 vaccine. No further RCVS attacks were reported. These findings indicate that the COVID-19 vaccine might induce RCVS attacks in susceptible individuals, and targeting the angiotensin 2 receptor could be a preventive option.

## 1. Background

Reversible cerebral vasoconstriction syndrome (RCVS) is an acute thunderclap headache, mimicking that of a ruptured aneurysm, caused by the narrowing of cerebral vessels. The main manifestation of RCVS is a headache lasting anywhere between a few minutes and several days with or without additional neurological symptoms [1]. A single RCVS attack is possible, but, usually, patients have a mean of four attacks with 1–4 weeks of moderate headaches between exacerbations. Attacks can be triggered by stress, coughing, sexual intercourse, and exertion [2]. RCVS occurs in all age groups, but occurrences peak around 42 years and are more often in women [1]. Although the true incidence of RCVS is unknown, RCVS is the most frequent cause of thunderclap headache and should be considered as a differential diagnosis in younger patients presenting with an acute headache to avoid underdiagnosis [3]. RCVS can occur spontaneously or can be secondary due to exogenous factors, typically vasoactive drugs. The exact underlying pathophysiology of RCVS is yet to be elucidated; several vasoactive biochemical and immunologic factors, including prostaglandins, catecholamines, nitric oxide, serotonin, and enothelin-1, are thought to be involved in the pathogenesis [2]. The diagnosis of RCVS is based on criteria provided in the third edition of the International Classification of Headache Disorders (ICHD-3) [3]. Computer tomography (CT) or magnetic resonance (MR) cerebral angiography can be normal during the first week after clinical onset but often show caliber irregularities affecting the anterior and posterior circulation bilaterally [1]. RCVS is self-limiting in 1–3 months, with the disappearance of the arterial abnormalities (hence “reversible”), and calcium channel blockers, typically nimodipine, can be used to treat RCVS attacks with a good prognosis [2].

## 2. Case Presentation

A 30-year-old male, known to have RCVS and bipolar disorder, presented to the department of neurology with three new attacks mimicking previous RCVS attacks and fulfilling the diagnostic criteria for RCVS after receiving the first Pfizer COVID-19 vaccine.

The patient was diagnosed with RCVS three years previously after a long hospitalization with suspected subarachnoid hemorrhage. The CT-angiography in 2018 showed no hemorrhage but segmental cerebral vasoconstriction of the basilar artery (Figure 1). A week later, the vasoconstriction could not be reidentified on a new CT-angiography. He started nimodipine treatment and has been taking it since.

The present-day symptoms started 12 h after COVID-19 vaccination with a thunderclap headache at night followed by a pressing headache characterized as 7 on the numerical rating scale (NRS). A new thunderclap headache arose the following evening during sexual intercourse and decreased to a moderate headache lasting 3 h. The next morning, a third episode of thunderclap headache began spontaneously, followed by nausea, dizziness, and a pressing headache (NRS 4–5) localized to the right temporal lobe. The patient described the symptoms as identical to the symptoms of previous RCVS attacks but with much higher frequency than usual. At the examination 48 h after the COVID-19 vaccination and 36 h since the first symptoms started, the patient had no dizziness and displayed no visual, speaking, or motoric disturbances. He had a completely normal neurological exam, blood samples, vital parameters, electrocardiogram (ECG), and computed tomography cerebrum (CTC) (Figure 1). The patient denied drug abuse and earlier severe acute respiratory syndrome coronavirus 2 (SARS-CoV-2) infection, and the SARS-CoV-2 polymerase chain reaction (PCR) test was negative.

To prevent further attacks of RCVS related to the Pfizer COVID-19 vaccine, the patient was prescribed 50 mg losartan one time daily until the follow-up at the Danish Headache Center two weeks later. Losartan treatment continued until three days after the second vaccine, and the patient experienced no more RCVS attacks.

## 3. Discussion

Cerebrovascular complications arise in 0.5–5% of acute SARS-CoV-2 infections, including ischemic stroke, intracerebral hemorrhage, and cerebral venous sinus thrombosis [4]. Several cases of RCVS attacks after SARS-CoV-2 infection have been reported. In a multicenter case series of ten patients with concurrent RCVS and SARS-CoV-2 infections, nine patients were diagnosed with COVID-19 within 30 days before the RCVS diagnosis. Five patients experienced at least one thunderclap headache with associated neurologic deficits, including encephalopathy, hemiparesis, aphasia, and visual deficits. Three patients did not have a provoking illness or use of vasoactive agents [4]. Two additional case reports described COVID-19 patients presenting with RCVS attacks confirmed by CT angiography, and one patient was treated with nimodipine and the other with verapamil [5,6].

A recently published case report described a patient diagnosed with RCVS 18 days after the second shot of Moderna COVID-19 vaccination. The patient presented with sudden onset of blurred vision bilaterally accompanied by a focal headache over the right occipital lobe. The morning after, a thunderclap headache arose after sneezing with a recurrence of visual impairment. Cerebral MR showed an acute cortical ischemic lesion in the territory of the right posterior cerebral artery, and MR angiography revealed discontinuation of the right P1 segment of the posterior cerebral artery that resolved after seven days of nimodipine treatment [7]. This finding further supports that COVID-19 and its vaccines might cause RCVS attacks [4,5,6,7].

Angiotensin-converting enzyme 2 (ACE-2) is suspected to be a key component of the pathophysiological mechanism underlying SARS-CoV-2 infection-induced RCVS [8]. ACE-2 is a membrane protein that is expressed in several organs, including the heart, lungs, blood vessels, and gastrointestinal tract. It is part of the renin-angiotensin-aldosterone system (RAAS), and its primary function is to downregulate the vasoconstrictive peptide angiotensin 2 to the vasodilative peptides angiotensin 1–7 (Figure 2) [9].

Several studies have shown that SARS-CoV-2 spike protein interacts with ACE-2 to promote cellular entry and initiate infection. This leads to the downregulation of ACE-2 and, consequently, higher levels of angiotensin 2, inducing vasoconstriction and possibly causing an RCVS attack (Figure 3). The Pfizer COVID-19 vaccine consists of stabilized mRNA encoding the spike protein with two amino acid replacements that maintain the spike protein at the prefusion state [10], which promotes cell signaling and elicits virus-neutralizing antibodies [8]. It is therefore conceivable that, after mRNA delivery into human cells, the vaccine-expressed spike protein, either membrane-bound or free-floating (released by vaccinated dead cells), might similarly interact with ACE-2, resulting in the upregulation of angiotensin 2. However, this hypothesis still lacks preclinical evidence. The fact that losartan, an angiotensin 2 inhibitor, prevented new attacks strengthens the notion that RCVS attacks in the patient were induced by the mRNA-based vaccine.

Another explanation could be that Atacand (candesartancilexetil), also an angiotensin 2 inhibitor, is used as prophylaxis in difficult-to-treat migraine patients [12]. The mechanism of action in reducing migraine is yet unknown. Therefore, the supposed effect of losartan on the prevention of RCVS could be due to the general prophylactic effect of angiotensin 2 inhibitors on headaches rather than the RAAS-mediated vasoconstriction. 

The lack of RCVS attacks after the second COVID-19 vaccination could also be confounded by the statistical phenomenon “regression to the mean”. This refers to the tendency of results that are extreme by chance on preliminary tests to move closer to the average in subsequent measurements. This statistical phenomenon can lead to the false conclusion that the intervention caused the effect. However, the fact that there have been several examples of RCVS cases after COVID-19 infection makes this explanation unlikely.

The prophylactic medication for RCVS patients (nimodipine) causes vasodilation, much like losartan. Why could nimodipine alone not prevent RCVS attacks after COVID-19 vaccination? Nimodipine is a calcium channel antagonist characterized by its ability to cross the blood-brain barrier. It blocks voltage-gated L-type calcium channels in vascular smooth muscle cells inhibiting calcium influx and subsequently inducing vasoconstriction [13]. This is an entirely different mechanism of action. Targeting different receptors on cerebral arteries with the combination of losartan and nimodipine might have resulted in more effective vasodilation, consequently preventing more RCVS attacks.

## 4. Conclusions

The present case report shows that the Pfizer COVID-19 vaccination can be followed by RCVS attacks in a patient with a history of RCVS. This suggests a possible association between COVID-19 vaccination and RCVS attacks in susceptible individuals, and targeting the angiotensin 2 receptor could be a preventive option. The present case adds to the very limited literature about a new disease (COVID-19) and its vaccination which is in the process of unfolding. More cases are needed to establish whether there is an association and whether it is a causal relationship.

### Diagnostic Criteria for RCVS According to ICHD-3

1.
**Acute headache attributed to reversible cerebral vasoconstriction syndrome (RCVS)**
Description: Headache caused by reversible cerebral vasoconstriction syndrome (RCVS), typically thunderclap headache recurring over one to two weeks, often triggered by sexual activity, exertion, Valsalva maneuvers and/or emotion. Headache can remain the sole symptom of RCVS or be a warning symptom preceding hemorrhagic or ischemic stroke.A.Any new headache fulfilling criterion CB.Reversible cerebral vasoconstriction syndrome (RCVS) has been diagnosedC.Evidence of causation demonstrated by either or both of the following:1.Headache, with or without focal deficits and/or seizures, has led to angiography (with a “string of beads” appearance) and diagnosis of RCVS2.Headache has one or more of the following characteristics:(a)Thunderclap onset(b)Triggered by sexual activity, exertion, Valsalva maneuvers, emotion, bathing and/or showering(c)Present or recurrent during ≥1 month after onset, with no new significant headache after >1 monthD.Either of the following:1.Headache has resolved within three months of onset2.Headache has not yet resolved but three months from onset have not yet passed
E.Not better accounted for by another ICHD-3 diagnosis
2.
**Acute headache probably attributed to reversible cerebral vasoconstriction syndrome (RCVS)**
Description: Headache typical for reversible cerebral vasoconstriction syndrome (RCVS), namely thunderclap headache, recurring over one to two weeks and triggered by sexual activity, exertion, Valsalva manœuvres and/or emotion, but the intracranial arterial beading typical of RCVS has not been demonstrated by cerebral angiography A.Any new headache fulfilling criterion CB.Reversible cerebral vasoconstriction syndrome (RCVS) is suspected, but cerebral angiography is normalC.Probability of causation demonstrated by all of the following 1.At least two headaches within one month, with all three of the following characteristics(a)Thunderclap onset and peaking in <1 min(b)Severe intensity(c)Lasting ≥ 5 min2.At least one thunderclap headache has been triggered by one of the following
(a)Sexual activity (just before or at orgasm)(b)Exertion(c)Valsalva-like maneuver(d)Emotion(e)Bathing and/or showering(f)Bending
3.No new thunderclap or other significant headache occurs >1 month after onsetD.Either of the following1.Headache has resolved within three months of its onset2.Headache has not yet resolved, but three months from its onset have not yet passedE.Not better accounted for by another ICHD-3 diagnosis.
3.
**Persistent headache attributed to past reversible cerebral vasoconstriction syndrome (RCVS)**
Description: Headache caused by reversible cerebral vasoconstriction syndrome (RCVS) and persisting for more than three months after onset.A.Headache previously diagnosed as 6.7.3.1 Acute headache attributed to reversible cerebral vasoconstriction syndrome (RCVS) and fulfilling criterion CB.Normalization of cerebral arteries, shown by follow-up indirect or direct angiography, within three months of onset of RCVSC.Headache has persisted for >3 months after its onsetD.Not better accounted for by another ICHD-3 diagnosis [3].

## Figures and Tables

**Figure 1 vaccines-10-00823-f001:**
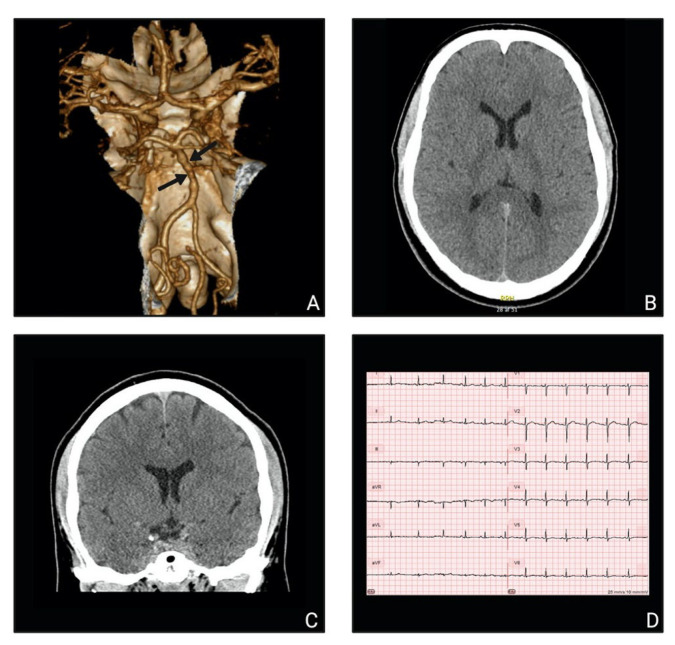
Paraclinical investigations: (**A**) CT angiography three-dimensional reconstruction from 2018 showing alternating segments of the basilar artery with constriction and dilation (“sausage on a string” appearance); (**B**,**C**) normal CTC-scans from 2021 in the transversal plane (**B**) and the coronal plane (**C**); (**D**) normal ECG from 2021.

**Figure 2 vaccines-10-00823-f002:**
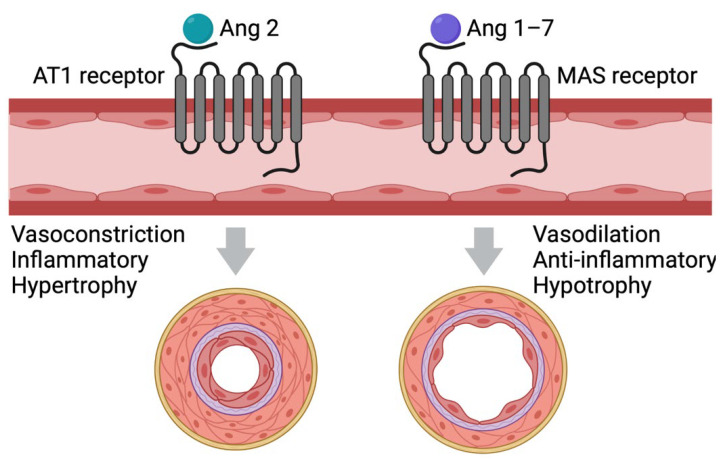
**Effect of angiotensin 2 and angiotensin 1–7 on the vasculature.** Angiotensin 2 binds to the G-protein coupled AT1-receptor, activating a downstream signaling pathway, leading to vasoconstriction, hypertropia, and inflammation of the blood vessels. Angiotensin 1–7 binds to the G-protein coupled MAS-receptor, which counter-regulates the effect of angiotensin 2. AT1: Angiotensin 2 Type 1; MAS: mitochondrial assembly.

**Figure 3 vaccines-10-00823-f003:**
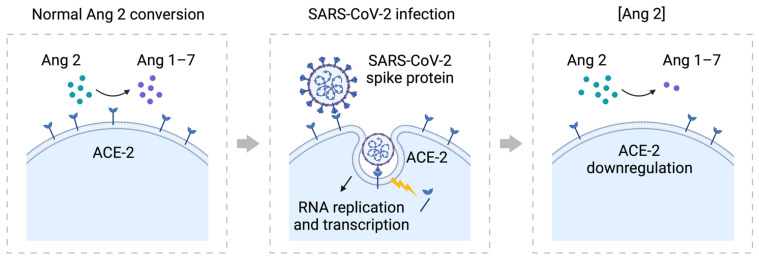
**The role of SARS-CoV-2 spike protein in the downregulation of ACE-2**. ACE-2 converts angiotensin 2 to angiotensin 1–7 as part of the RAAS system. The SARS-CoV-2 spike protein binds to the ACE-2 receptor to fuse with the cell and start RNA replication and transcription. This leads to the downregulation of ACE-2, an increase in angiotensin 2, a decrease in angiotensin 1–7, and eventually vasoconstriction [8,10,11].

## Data Availability

Not applicable.

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
