# Peer review of "COVID-19 Vaccination Might Induce Reversible Cerebral Vasoconstriction Syndrome Attacks: A Case Report"

_vaccines, 2022, doi:10.3390/vaccines10050823_

Round 1
Reviewer 1 Report
This case report adds to the current literature regarding Reversible cerebral vasoconstriction syndrome (RCVS).
This is an interesting report, however, the presentation and description of the case could and should definitely be improved.
Starting with the background please provide more information regarding the Reversible cerebral vasoconstriction syndrome (RCVS), incidence, pathophysiology, etc.
The case was presented extremely concisely, it would help to provide more information that dissipates possible confounders.
The discussion should expand more on the direct comparison and discussion with the already reported possible cases of RCVS in relation to the COVID-19 vaccine, what are the differences in terms of presentation, treatments, preventions in previous cases, compared to the case presented here.
Minor grammatical errors in the manuscript, please revise accordingly.
Author Response
Comments from reviewer 1
This case report adds to the current literature regarding Reversible cerebral vasoconstriction syndrome (RCVS). This is an interesting report, however, the presentation and description of the case could and should definitely be improved.
- Starting with the background please provide more information regarding the Reversible cerebral vasoconstriction syndrome (RCVS), incidence, pathophysiology, etc.
Response: Thank you very much for your comments. More information on pathogenesis, incidence, and pathophysiology would be a good supplement to the background. This has been added (added on page 1, line 19-43).
- The case was presented extremely concisely, it would help to provide more information that dissipates possible confounders.
Response: We have added details about the time of the vaccination, when the first symptoms started, and when the patient was admitted to the hospital. Furthermore, vital parameters were normal, the patient denied drug abuse and earlier SARS-CoV-2 infection. ECG and CTC are provided as a new figure (added on page 2 line 53-67, figure 1 on page 2 line 71-75).
- The discussion should expand more on the direct comparison and discussion with the already reported possible cases of RCVS in relation to the COVID-19 vaccine, what are the differences in terms of presentation, treatments, preventions in previous cases, compared to the case presented here.
Response: More information on the terms of presentation and treatments have been added. We also included a recently published case of RCVS after the Moderna COVID-19 vaccine in the discussion (added page 3 line 78-96).
- Minor grammatical errors in the manuscript, please revise accordingly.
Response: Thank you for your comment. These have been revised.
Reviewer 2 Report
The authors reported A 30-year-old male previously diagnosed with reversible cerebral vasoconstriction syndrome (RCVS), presented to the department of neurology with an accumulation of RCVS attacks after 8 receiving the first Pfizer COVID-19 vaccine. I have some comments about the case report to be addressed by the authors.
- In the title, replace RCVS with Reversible Cerebral Vasoconstriction Syndrome.
- Include the electrocardiogram (ECG), and computer tomography (CT) of the head for the discussed case in the result section or even as a supplementary file.
Author Response
Comments from reviewer 2
The authors reported A 30-year-old male previously diagnosed with reversible cerebral vasoconstriction syndrome (RCVS), presented to the department of neurology with an accumulation of RCVS attacks after receiving the first Pfizer COVID-19 vaccine. I have some comments about the case report to be addressed by the authors.
- In the title, replace RCVS with Reversible Cerebral Vasoconstriction Syndrome.
Response: Thank you very much for your comments. This has been changed (page 1 line 1-3).
- Include the electrocardiogram (ECG), and computer tomography (CT) of the head for the discussed case in the result section or even as a supplementary file.
Response: These have been added, see figure 1 (page 2 line 71-75).
Reviewer 3 Report
Lund and Al-Karagholi describe in this case study a patient with a history of RCVS who presented with thunderclap headache starting 12 hours after vaccination with the first dose of an mRNA vaccination with BNT162b2 (Biontech/Pfizer). They claim these thunderclap headaches to be caused by a new RCVS attack, which is induced by the vaccination. In the manuscript the diagnosis of the current RCVS attack is solely supported by the RCVS history of the patient and self-reported thunderclap headache, however, no alterations in the CT etc. are seen. The conclusion that COVID-19 vaccination induces RCVS attacks based on these data and the fact that some RCVS attacks were previously reported in the literature after SARS-CoV-2 infection seems to be a bit far fetched.
- The authors claim in the title and the last sentence of the abstract (line 13/14) that COVID vaccination induces RCVS attacks. These claims seem to over-interpret the data they present. They present a single patient with a history of RCVS attacks who presents after the first dose of vaccination with an mRNA vaccine with headache (3 episodes of thunderclap headache). However, the examination of the patient (not clear from the manuscript when after the vaccination/episodes of thunderclap headache the patient has been examined) revealed no abnormalities. Was the diagnosis RCVS confirmed other than by the patients description of the headache?
- The authors claim in the introduction that “to diagnose RCVS a computer tomography (CT) 27 or a magnetic resonance (MR) cerebral angiography is needed to show calibre irregularities affecting the anterior and posterior circulation bilaterally” (lines 27-29). However, for the here presented patient CT was normal as all other examinations conducted. How was the diagnosis of the patient confirmed.
- Lines 48/49 say that the patient was not previously infected with COVID-19. COVID-19 is the disease and the infecting virus. Therefore, this should be corrected to SARS-CoV-2 infection. Additionally, the authors should add the information if this was self-reported only or if they confirmed this by serology.
- Line62-66:
“These findings suggested a possible association between COVID-19 infection and RCVS attacks.
Angiotensin-converting-enzyme 2 (ACE-2) is suspected to be a key component of the pathophysiological mechanism underlying COVID-19 vaccine induced RCVS.”
The authors cite paper showing a possible connection of SARS-CoV-2 infection with RCVS attacks and conclude from this that the ACE2 receptor is responsible for vaccine induced RCVS. To draw this conclusion they would first need to establish that CODID-19 vaccines could cause RCVS. Is there any evidence from the literature that vaccination could cause RCVS?
- Both figures are not clear and why they are shown. Figure 2 for example mixes infection and vaccination. The middle panel shows infection of a cell with virus, however, the panel is entitled with Pfizer Covid vaccination. The picture shown has nothing to do with vaccination.
- Lines 101-103: The conclusion does not seem to be clearly supported by the data presented.
Minor:
- “COVID” and “Covid” and “COVID-19” are all used. This should be harmonized.
Author Response
Comments from reviewer 3
Lund and Al-Karagholi describe in this case study a patient with a history of RCVS who presented with thunderclap headache starting 12 hours after vaccination with the first dose of an mRNA vaccination with BNT162b2 (Biontech/Pfizer). They claim these thunderclap headaches to be caused by a new RCVS attack, which is induced by the vaccination. In the manuscript the diagnosis of the current RCVS attack is solely supported by the RCVS history of the patient and self-reported thunderclap headache, however, no alterations in the CT etc. are seen. The conclusion that COVID-19 vaccination induces RCVS attacks based on these data and the fact that some RCVS attacks were previously reported in the literature after SARS-CoV-2 infection seems to be a bit far fetched.
- The authors claim in the title and the last sentence of the abstract (line 13/14) that COVID vaccination induces RCVS attacks. These claims seem to over-interpret the data they present. They present a single patient with a history of RCVS attacks who presents after the first dose of vaccination with an mRNA vaccine with headache (3 episodes of thunderclap headache). However, the examination of the patient (not clear from the manuscript when after the vaccination/episodes of thunderclap headache the patient has been examined) revealed no abnormalities. Was the diagnosis RCVS confirmed other than by the patients description of the headache?
Response: Thank you very much for this question and your other comments. The patient was diagnosed with RCVS three years previously (2018) where the CT-angiography showed segmental cerebral vasoconstriction of the basilar artery. Shortly after COVID-19 vaccination, the patients experienced several RCVS attacks mimicking previous attacks and fulfilling the diagnostic criteria for RCVS provided in the third edition of the International Classification of Headache Disorders (ICHD-3). Moreover, MR-, CT- and even catheter-angiography can be normal during the first week after clinical onset (added page 1 line 8-16, page 1 line 39-40, and page 2 line 48-51).
- The authors claim in the introduction that “to diagnose RCVS a computer tomography (CT) 27 or a magnetic resonance (MR) cerebral angiography is needed to show calibre irregularities affecting the anterior and posterior circulation bilaterally” (lines 27-29). However, for the here presented patient CT was normal as all other examinations conducted. How was the diagnosis of the patient confirmed.
Response: Thank you very much for this question. This has been taken care of in previous comments.
- Lines 48/49 say that the patient was not previously infected with COVID-19. COVID-19 is the disease and the infecting virus. Therefore, this should be corrected to SARS-CoV-2 infection. Additionally, the authors should add the information if this was self-reported only or if they confirmed this by serology.
Response: We agree that it should be corrected to SARS-CoV-2 infection. COVID-19 antibody test was negative (added page 2, line 64-66).
- Line62-66:
“These findings suggested a possible association between COVID-19 infection and RCVS attacks. Angiotensin-converting-enzyme 2 (ACE-2) is suspected to be a key component of the pathophysiological mechanism underlying COVID-19 vaccine induced RCVS.”
The authors cite paper showing a possible connection of SARS-CoV-2 infection with RCVS attacks and conclude from this that the ACE2 receptor is responsible for vaccine induced RCVS. To draw this conclusion they would first need to establish that COVID-19 vaccines could cause RCVS. Is there any evidence from the literature that vaccination could cause RCVS?
Response: Thank you for this interesting question. There is, to our knowledge, only one case reporting RCVS-attacks after COVID-19 vaccination. This has been added to the case report. The other papers cited are RCVS-diagnosis after SARS-CoV-2 infection. We compare Pfizer’s COVID-19 vaccination with SARS-CoV-2 infection because the mechanism of cell entry is similar (with spike protein and ACE-2). The present case adds to the very limited literature about a new disease (COVID-19) and its vaccination which is in the process of unfolding (added to page 3 and 4 line 88-118).
- Both figures are not clear and why they are shown. Figure 2 for example mixes infection and vaccination. The middle panel shows infection of a cell with virus, however, the panel is entitled with Pfizer Covid vaccination. The picture shown has nothing to do with vaccination.
Response: We agree that figure 2 (now called figure 3) mixes Pfizer vaccine with SARS-CoV-2 infection. This has been corrected. Our aim with the figure is to present the mechanism that SARS-CoV-2 enters the cell and downregulates ACE-2, leading to vasoconstriction and possibly RCVS-attacks. mRNA-based vaccines like Pfizer use the same mechanism and we suggest that this could also explain vaccine induced RCVS-attacks. Figure 2 illustrates how angiotensin 2 leads to vasoconstriction. It does not show how SARS-CoV-2 affects angiotensin 2, which is why we also have figure 3 showing this (added page 4 line 115-127).
- Lines 101-103: The conclusion does not seem to be clearly supported by the data presented.
Response: Our conclusion states that we suggest a possible association between COVID-19 vaccination and RCVS attacks, but more cases are needed to establish a causal relationship (added page 5 line 148-155).
- Minor: “COVID” and “Covid” and “COVID-19” are all used. This should be harmonized.
Response: All mentioning of the vaccine is now mentioned as “COVID-19 vaccine”. This is seen in the title and throughout the entire text.
Round 2
Reviewer 1 Report
The authors have successfully addressed the comments raised by the reviewer and have significantly improved the quality, content, and presentation of the manuscript.
Author Response
Thank for your comments.
Reviewer 3 Report
Thank you for addressing my comments. There are still a few questions open.
- Line 63/64: I suggest to change COVID-19 antibody test to SARS-CoV-2 antibody test. Additionally it would be helpful to add which test from which company has been performed.
- Line 109-112 and 120/121: The mRNA vaccine should not use the same entry way as natural SARS-CoV-2. The later fuses with the cell as consequence of binding of spike to the ACE2 receptor. The vaccine should fuse directly with its lipid envelope to the cell membrane. To my knowledge the mRNA lipid nanoparticle does not contain spike as protein in its envelope but just as coding information in the mRNA. Additionally, the two prolin substitutions in the mRNA vaccines stabilize the spike in the pre-fusion conformation and therefore should anyhow render it impossibible to fuse with the cell membrane. The cell will produce membrane-bound spike (in a fusion incompetent form) following uptake of the vaccine lipid nanoparticles. If this can lead to downregulation of ACE2 in the muscle is not clear, a systemic ACE2 downregulation upon intramucular immunization seems to be highly unlikely.
- Although a connection of the RCVS attack described in this case study with the COVID-19 vaccination might exist, the explaination model presented here is not completely plausible. Infection and mRNA vaccination follow different mechanisms of cell entry and the quality of the produced spike is different, fusogenic for infection and pre-fusion stabilized for mRNA vaccination. Additionally, spike production occurs at different body compartments, for vaccination within the muscle, for infection primarily in the respiratory tract but also invades other organs.
Author Response
- Line 63/64: I suggest to change COVID-19 antibody test to SARS-CoV-2 antibody test. Additionally it would be helpful to add which test from which company has been performed.
Response: Thank you for your comments. This has addressed on page 2, line 64-65.
- Line 109-112 and 120/121: The mRNA vaccine should not use the same entry way as natural SARS-CoV-2. The later fuses with the cell as consequence of binding of spike to the ACE2 receptor. The vaccine should fuse directly with its lipid envelope to the cell membrane. To my knowledge the mRNA lipid nanoparticle does not contain spike as protein in its envelope but just as coding information in the mRNA. Additionally, the two prolin substitutions in the mRNA vaccines stabilize the spike in the pre-fusion conformation and therefore should anyhow render it impossibible to fuse with the cell membrane. The cell will produce membrane-bound spike (in a fusion incompetent form) following uptake of the vaccine lipid nanoparticles. If this can lead to downregulation of ACE2 in the muscle is not clear, a systemic ACE2 downregulation upon intramucular immunization seems to be highly unlikely. Although a connection of the RCVS attack described in this case study with the COVID-19 vaccination might exist, the explaination model presented here is not completely plausible. Infection and mRNA vaccination follow different mechanisms of cell entry and the quality of the produced spike is different, fusogenic for infection and pre-fusion stabilized for mRNA vaccination. Additionally, spike production occurs at different body compartments, for vaccination within the muscle, for infection primarily in the respiratory tract but also invades other organs.
Response: This has been addressed on page 4, line 111-128.